# Comparison of Functional Outcomes between Supervised Rehabilitation and Telerehabilitation in Female Patients with Patellofemoral Pain Syndrome during the COVID-19 Pandemic

**DOI:** 10.3390/ijerph20032233

**Published:** 2023-01-26

**Authors:** Jin Hyuck Lee, Ki Hun Shin, Gyu Bin Lee, Seiwook Son, Ki-Mo Jang

**Affiliations:** 1Department of Sports Medical Center, Korea University College of Medicine, Seoul 02841, Republic of Korea; 2Department of Orthopaedic Surgery, Korea University College of Medicine, Seoul 02841, Republic of Korea

**Keywords:** knee joint, patellofemoral pain syndrome, COVID-19, telerehabilitation, supervised rehabilitation

## Abstract

Patellofemoral pain syndrome (PFPS) is a common cause of anterior knee pain, and therapeutic exercises are recommended. During the COVID-19 pandemic, despite recommendations on the importance of telerehabilitation, insufficient studies have investigated functional outcomes between supervised rehabilitation and telerehabilitation in patients with PFPS. This study aimed to compare the muscle strength, muscle activation time, and patient-reported outcomes between supervised rehabilitation and telerehabilitation in female patients with PFPS. A total of 61 patients (supervised, *n* = 30; telerehabilitation, *n* = 31) participated. Muscle strength and activation time of the quadriceps and hamstrings were measured using an isokinetic device. Hip muscle strength was evaluated using a hand-held dynamometer. Patient-reported outcomes were measured using the visual analog scale (VAS) for pain, Kujala Anterior Knee Pain Scale (AKPS) for functional ability, and Tampa scale for kinesiophobia (TSK-11). No significant differences were found in muscle strength, muscle activation time, or patient-reported outcomes of the involved knees between the two groups (*p* > 0.05). In addition, the rate of change in all parameters did not significantly differ between the two groups (*p* > 0.05). Telerehabilitation, such as a home-exercise program supervised by physical therapists, may be as effective as supervised rehabilitation in improving functional outcomes in female patients with PFPS.

## 1. Introduction

Patellofemoral pain syndrome (PFPS) is commonly described as pain around the patella and complaints of aggravated knee pain from jumping, running, kneeling, long hours of sitting, and climbing up or down stairs [1]. PFPS may be caused by various factors, such as quadriceps and hip muscle weakness, muscle imbalance and inflexibility, foot and ankle postures, prolonged muscle activation, and lack of proprioception and neuromuscular control [2,3]. Hence, therapeutic exercise treatment may be recommended in preference to surgical treatment [4,5,6].

Telerehabilitation (tele-rehab) was developed as a home treatment after acute care [7,8] and has been further developed as a treatment method by telecommunication to compensate for the traditional face-to-face treatment method. In 2019, the spread of severe acute respiratory syndrome coronavirus 2 started, and governments around the world began to propose several systems, such as quarantine, occupancy restrictions, and social distancing. Hence, several recent studies have compared functional outcomes in supervised rehabilitation (supervised-rehab) and tele-rehab in response to the coronavirus disease 2019 (COVID-19) outbreak [9,10,11]. In this situation, tele-rehab is strongly recommended because it has the benefits of less time and lower costs [9,10]. In recent studies, tele-rehab has been proven effective in improving functional outcomes in patients with hip arthroplasty [9], knee osteoarthritis (OA) [12], and low back pain [10]. However, to date, insufficient studies have investigated functional outcomes in terms of muscle strength, muscle activation time, and patient-reported outcomes (including the visual analog scale [VAS], Kujala Anterior Knee Pain Scale (AKPS), and Tampa scale for kinesiophobia [TSK-11]) between supervised-rehab and tele-rehab in patients with PFPS.

During the COVID-19 pandemic, our institution also offered treatment with social distancing, such as limiting the number of visiting patients. Therefore, the study aimed to compare the muscle strength of the quadriceps, hamstring, and hip muscles, muscle activation time of the quadriceps and hamstring muscles, and patient-reported outcomes between supervised-rehab and tele-rehab in female patients with PFPS. We hypothesized that tele-rehab would be as effective as supervised-rehab in improving functional outcomes in female patients with PFPS.

## 2. Materials and Methods

### 2.1. Study Subjects

This study complied with the Declaration of Helsinki and informed consent was obtained from all patients or their legal guardians. This prospective comparative study was approved by the institutional review board of our institution (No. 2017AN0830), and 72 female patients with anterior knee pain were consecutively recruited between February 2019 and January 2020. Seventy-two female patients diagnosed with PFPS by an orthopedic surgeon were identified through physical examination, medical record reviews, plain radiography, and magnetic resonance imaging to confirm abnormal patellofemoral bony structures and cartilage lesions. For a diagnosis of PFPS, patients needed to have at least two anterior or retropatellar knee pain during functional activities such as running, jumping, kneeling, long hours of sitting, and climbing up or down stairs [1]. In this study, only female patients with PFPS who had a high training adherence rate >80% were included. A training adherence rate >80% was clinically meaningful to facilitate the effectiveness of the intervention [13,14]. Therefore, based on a previous study [15], the adherence rate was measured by the total number of sessions performed by the participant divided by the total number of treatment sessions. We excluded 11 patients for the following reasons: training adherence rate <80%, chondromalacia, OA, prior knee surgery, and spine and vestibular disorders [16,17]. In addition, patients who were unable to complete the isokinetic test with pain were excluded. Of the 72 female patients enrolled (Figure 1), 11 were excluded and 61 were analyzed in the present study; they were divided into the supervised-rehab (*n* = 30) and tele-rehab (*n* = 31) groups.

### 2.2. Isokinetic Knee Muscle Strength and Muscle Activation Time

The muscle strength and activation time of the quadriceps and hamstring muscles were measured using an isokinetic device (Biodex Multi-Joint System 4, Biodex Medical Systems, Inc., Shirley, NY, USA). In the sitting position with 90° flexion of the hip and knee joints, the lateral femoral condyle of the knee joint was aligned with the rotational center of an isokinetic dynamometer and then evaluated for hamstring and quadriceps muscle strength with 90° knee flexion angle and 0° knee extension angle, respectively. Before starting this test, a warm-up set was performed with five repetitions of knee extension/flexion at a sub-maximal intensity at 180°/s. Muscle strength was measured by peak torque during five maximal repetitions of flexion and extension motions at 180°/s, and peak torque normalized to the body weight (peak torque/body weight, N·m·kg^−1^ × 100) was used to evaluate muscle strength [18,19]. In this study, the intraclass correlation coefficients (ICCs) were 0.89 and 0.84 for the quadriceps and hamstring muscles, respectively. Muscle activation time was assessed using the acceleration time (AT, milliseconds), defined as the time it took for the pre-set angular velocity (180°/s in our study) during maximal contraction [18,19,20]. Acceleration and velocity can affect the arthrokinetic reflex, which is closely related to the mechanism of muscle activation [20,21]. Thus, a rapid AT indicates superior muscle activation responses, which can affect neuromuscular control, defined as unconscious activation between muscles and motor neurons to maintain and restore functional joint stability [22]. In the present study, the ICCs were 0.82 and 0.79 for the quadriceps and hamstring muscles, respectively.

### 2.3. Isometric Hip Muscle Strength

Isometric hip muscle strength was measured using a hand-held dynamometer (microFET2, Hoggan Scientific, LLC, Salt Lake City, UT, USA). Based on a previous study [23], for evaluating posterolateral hip complex strength such as hip abductor and external rotator [24], the hip stability isometric test (HipSIT) was performed with the leg facing up, while the participant maintained 45° flexion and 20° abduction of the hip joint and 90˚ flexion of the knee joint in a side-lying position. The HipSIT was performed twice in total, with a rest period of 30 s between measurements, and the average value was recorded. The measured muscle strength value (kilogram-force, kgf) was normalized to the weight of each participant (strength/body mass), and the minimal clinically important difference (MCID) was 0.036 kgf/kg [23]. In a previous study [23], intra-rater and inter-rater ICCs for the HipSIT were 0.981 and 0.981, respectively. In the present study, the ICC was 0.946. Mentiplay et al. [25]. reported that isometric hip muscle strength showed moderate to excellent validity between a hand-held and isokinetic dynamometry.

### 2.4. Patient-Reported Outcomes

Isokinetic VAS and AKPS were used to evaluate pain and knee function, respectively [18,19]. VAS was recorded as the worst pain during active movement. A score of 0 indicated no pain, and a score of 10 indicated worst pain. The AKPS consists of 13 questions and can be evaluated on a scale of 0–100. A lower score indicates greater discomfort and disability. Based on a previous study in patients with PFPS [26], the MCID for the VAS and AKPS were 1.5–2.0 points and 8–10 points, respectively, and the ICCs were 0.88 and 0.81 for the VAS and AKPS, respectively. The TSK-11 was used to assess the fear of reinjury or movement. According to a previous study [27], fear of pain or movement, rather than the pain itself, may have a greater influence on the disability. The TSK-11 consists of 11 questions, with total scores ranging from 11 to 44. The higher the score, the greater the fear of reinjury or movement. In a previous study, the MCID was 4 points [28] and the ICC was 0.64–0.91 [29].

### 2.5. Conservative Rehabilitation Protocol and Interventions

All participants followed the same rehabilitation protocol for both knees. The exercise program was performed three times a week for 6 weeks [30], which aimed to improve flexibility, strength, proprioception, and neuromuscular control, and consisted of open and closed kinetic chain (CKC) exercises (Appendix A). The exercise program included the stretching, strengthening of the hip, knee, and core muscles, and balance exercises. (1) The supervised-rehab group visited our institution and performed an exercise program for 50 min, three times a week. The exercise program was performed by the same physical therapists, and the exercise intensity was determined by the evaluation and discretion of the physical therapists according to the patient’s symptoms. (2) The tele-rehab group was educated about the exercise program once and received a brochure with pictures and videos of the exercise program. The participants were instructed to execute the exercise interventions for 50 min, three times a week. The physical therapists in charge provided counseling and guidance on exercise progress, maintaining daily activities, and symptom improvement through text messages and phone calls, three times a week. If needed, physical therapists consulted on the exercise program through video calls. The physical therapists monitored adherence to the home exercise program.

### 2.6. Statistical Analysis

Based on a previous study on quadriceps strength in patients with PFPS [18,19], a difference >10% in quadriceps strength was regarded as a clinical difference between the PFPS patient group. A priori power analysis was performed to determine the sample size at a power of 0.8 and α level of 0.05. The results of a pilot study involving five knees in each group indicated that each group should include sixteen knees to detect a significant difference in quadriceps strength of >10% between the two groups (Cohen’s d: 1.042). Therefore, in the present study, we recruited 30 female patients with PFPS for the supervised-rehab group and 31 female patients for the tele-rehab group. The power of this study was 0.813. All continuous variables are presented as mean ± standard deviation. Student’s *t*-test was used to compare the differences in the muscle strength of the quadriceps, hamstring, and hip muscles, muscle activation time for quadriceps and hamstring muscles, and patient-reported outcomes between the supervised-rehab and tele-rehab groups. The paired *t*-test was used to compare two related variables before and after the intervention in the involved knees of each patient in both groups. To determine whether a continuous variable followed a normal distribution, the Shapiro–Wilk test was used. Significance was defined as a *p*-value of less than 0.05. Statistical analysis was performed using IBM SPSS Statistics for Windows, Version 21.0 (IBM corp., Armonk, NY, USA).

## 3. Results

Table 1 shows the demographic data of the participants, and no significant differences were found in sex, age, height, weight, and body mass index, Tegner activity scale score, and duration of injury (*p* > 0.05).

### 3.1. Comparison of Muscle Strength and Muscle Activation Time between the Two Groups

There were no statistically significant differences in the muscle strength for the quadriceps, hamstring, and hip muscles between the supervised-rehab and tele-rehab groups (*p* > 0.05, Table 2). The AT for the quadriceps and hamstring muscles was not statistically significantly different between the supervised-rehab and tele-rehab groups (*p* > 0.05, Table 2). In addition, the rate of change in the muscle strength and reaction time also did not significantly differ between the two groups (*p* > 0.05, Figure 2).

### 3.2. Comparison of Patient-Reported Outcomes between the Two Groups

No significant differences were found in the VAS, AKPS, or TSK-11 scores between the two groups (*p* > 0.05, Table 3). In addition, the rate of change in the VAS, AKPS, and TSK-11 scores did not significantly differ between the two groups (*p* > 0.05, Figure 2).

### 3.3. Comparison of Muscle Strength, Muscle Activation Time, and Patient-Reported Outcomes in Each Group

In each group, there were significant improvements in the muscle strength of the quadriceps (supervised: *p* < 0.001, tele-rehab: *p* < 0.001), hamstring (supervised: *p* < 0.001, tele-rehab: *p* < 0.001), and hip muscles (supervised: *p* < 0.001, tele-rehab: *p* < 0.001,) in the involved knees post-intervention compared to that with pre-intervention (Table 2). The AT of the quadriceps (supervised: *p* < 0.001, tele-rehab: *p* < 0.001) and hamstring muscles (supervised: *p* < 0.001, tele-rehab: *p* < 0.001) was significantly improved in the involved knees post-intervention compared to those with pre-intervention in each group (Table 2). Patient-reported outcomes, including VAS (supervised: *p* < 0.001, tele-rehab: *p* < 0.001), AKPS (supervised: *p* < 0.001, tele-rehab: *p* < 0.001), and TSK-11 (supervised: *p* < 0.001, tele-rehab: *p* < 0.001), were significantly improved in the involved knees post-intervention compared to those with pre-intervention in each group (Table 3).

### 3.4. Correlations between Adherence Rate and Rate of Change in Muscle Strength, Muscle Activation Time, and Patient-Reported Outcomes

Correlations between adherence rate and rate of change in muscle strength, muscle activation time, and patient-reported outcomes are shown in Table 4. In the supervised-rehab group, there were significantly positive correlations between adherence rate and rate of change in quadriceps (r = 0.516, *p* = 0.002) and hip muscle strength (r = 0.494, *p* = 0.006), AT of the quadriceps (r = 0.432, *p* = 0.021), AKPA score (r = 0.650, *p* = 0.001), and TSK-11 score (r = 0.444, *p* = 0.014). In the tele-rehab group, there were significantly positive correlations between adherence rate and rate of change for quadriceps strength (r = 0.637, *p* = 0.001), AT of the quadriceps (r = 0.371, *p* = 0.040), VAS score (r = 0.368, *p* = 0.042), AKPA score (r = 0.456, *p* = 0.010), and TSK-11 score (r = 0.509, *p* = 0.003). However, there was no significant correlation between adherence rate and hamstring muscle strength in either group (*p* > 0.05).

## 4. Discussion

The most important result of this study was that the muscle strength of the quadriceps, hamstring, and hip muscles, muscle activation time of the quadriceps and hamstring muscles, and patient-reported outcomes did not significantly differ between the supervised-rehab and tele-rehab groups.

In this study, no significant differences were found in the muscle strength of the quadriceps, hamstring, and hip muscles between the two groups. Although the reason for this result is unclear, it may be explained by adherence to training. Muscle strength can be improved by various factors such as nutrition, overloading weights, training status, and training adherence [31,32,33,34,35]. Among them, training adherence plays an important role in home-based exercise [31,34]. Specifically, a high level of training adherence may be the most important component in promoting the optimal effectiveness of an exercise program [13,36,37]. In a recent study, Hanson et al. [14] reported that high adherence to home-based exercise can improve muscle strength. In tele-rehab of the present study, it is believed that the physical therapists in charge increased the patient’s exercise adherence directly or indirectly through phone or video calls three times a week for 6 weeks. Therefore, tele-rehab managed by physical therapists may increase patient participation in exercise, resulting in increased patient exercise adherence; hence, there may be no difference in muscle strength between the supervised-rehab and tele-rehab groups because of the patient’s high exercise adherence, equal to that with supervised-rehab.

In the present study, the muscle activation time of the quadriceps and hamstring muscles did not show any significant differences between the two groups. A possible explanation for this result might be the use of the same rehabilitation protocol, such as dynamic stretching and CKC exercises with controlled lower extremity alignment. Dynamic stretching [18,38] and CKC [39,40] exercises may improve motor neurons, which may affect muscle activation [22,38,39]. Specifically, motor neurons activation may be improved by eccentric contraction [41] from dynamic stretching and CKC exercises. According to several previous studies [39,40,41], eccentric contraction may be more favorable for muscle activation than concentric contraction. Therefore, in the present study, both dynamic stretching and CKC exercises were performed and emphasized in both of the groups. Specifically, enhanced motor neuron and muscle activation may improve neuromuscular control [37,39,42,43], which may affect patient-reported outcomes [19,44]. Zech et al. [45] and Riemann and Lephart [22] reported that impaired neuromuscular function is considered the cause of persistent functional deficits, such as reduced maximal muscle strength, poor postural control, or prolonged muscle activation time. Therefore, persistent functional deficits lead to fear of movement [46,47], which may reduce training adherence [46], resulting in long-term mental, psychological, and functional deficits [48,49]. However, both supervised-rehab and tele-rehab may improve self-efficacy by increasing patients’ exercise participation and adherence [34,46,50]. In the present study, both muscle activation time and patient-reported outcomes (VAS, AKPS, and TSK-11) were significantly improved in both groups, which may explain the lack of differences in muscle activation time and patient-reported outcomes between the supervised-rehab and tele-rehab groups.

Azma et al. [12] reported no differences in patient-reported outcomes, such as VAS, Knee injury and Osteoarthritis Outcome Score (KOOS), and Western Ontario and McMaster Universities (WOMAC) scale score, in tele-rehab by a versed expert compared with those obtained with supervised-rehab in patients with knee OA. Albornoz-Cabello et al. [11] investigated patient-reported outcomes between using informative leaflets and tele-rehab managed by physical therapists in patients with PFPS. The authors found greater effectiveness in improving VAS, AKPS, and perception of neuropathic pain (DN4) in the tele-rehab managed by physical therapists than in the informative leaflet group. In addition, Rhim et al. [51] investigated patient-reported outcomes, such as VAS, KOOS, and TSK-11, between tele-rehab without the management of physical therapists (modeling video program vs. PowerPoint slides video) in patients with anterior cruciate ligament reconstruction. The authors found that patient-reported outcomes did not differ before and after interventions in either group. These findings indicate that tele-rehab managed by physical therapists may improve psychological and functional abilities [46].

This study has several limitations. First, no normal control group was included. Second, there may be a limit to generalizing the effect of tele-rehab owing to various factors such as age, physical activity, and sex, because only female patients with PFPS participated in our study [52]; thus, further studies are needed to clarify the results of this study. Finally, this study was a quasi-experimental study without randomization. Therefore, further high-quality studies with randomization and long-term follow-up are necessary to more clearly elucidate the results of this study.

## 5. Conclusions

Functional outcomes in both the supervised and telerehabilitation groups were significantly improved. Therefore, telerehabilitation, such as a home-exercise program supervised by physical therapists, may be as effective as supervised rehabilitation in improving functional outcomes in female patients with PFPS. Furthermore, telerehabilitation may be a suitable treatment method to improve psychological and functional ability during a pandemic, such as COVID-19. However, further studies are needed to determine whether a home-exercise program individually performed without the supervision of physical therapists is as effective as telerehabilitation.

## Figures and Tables

**Figure 1 ijerph-20-02233-f001:**
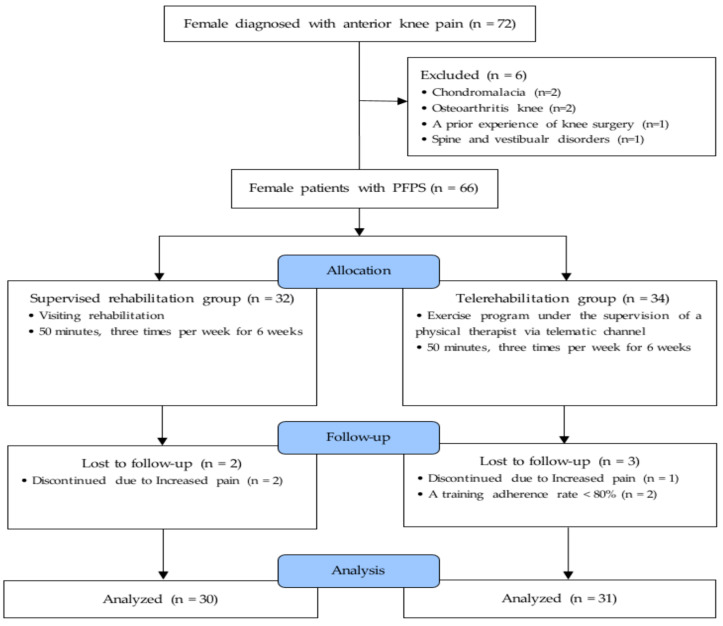
Study flow diagram.

**Figure 2 ijerph-20-02233-f002:**
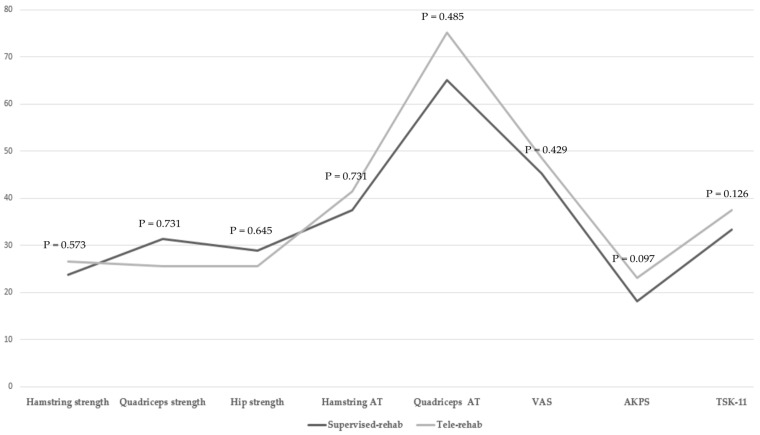
Comparison of the rate of change in muscle strength, muscle activation time, and patient-reported outcomes between the supervised rehabilitation group and telerehabilitation group.

**Table 1 ijerph-20-02233-t001:** Demographic data of participants in the supervised-rehab and tele-rehab groups.

	Supervised-Rehab Group (*n* = 30)	Tele-Rehab Group (*n* = 31)	*p*-Value
Sex (male/female)	0/30	0/31	
Age (years)	27.0 ± 2.39	26.96 ± 5.99	0.753
Height (cm)	160.49 ± 4.83	159.88 ± 6.11	0.670
Weight (kg)	56.64 ± 7.67	56.51 ± 11.56	0.788
Body mass index (BMI, kg/m^2^)	21.92 ± 2.76	22.04 ± 3.62	0.448
Pain duration (month)	26.14 ± 15.94	27.45 ± 22.46	0.753
Tegner activity scale	4.32 ± 0.82	4.25 ± 1.40	0.484
Adherence rate (%)	98.8 ± 0.3	98.5 ± 0.3	0.689
Injured side (right/left)	15/15	16/15	
Leg dominance	27/3	27/4	

Supervised-rehab, supervised rehabilitation; Tele-rehab, telerehabilitation. Note: The values are expressed as mean ± standard deviation.

**Table 2 ijerph-20-02233-t002:** Comparison of the muscle strength and muscle activation time in the supervised-rehab and tele-rehab groups.

		Supervised-Rehab Group	Tele-Rehab Group	MD (95% CI)	Effect Size	*p*-Value
Hamstring strength	Pre-intervention	71.6 ± 18.8	75.3 ± 19.8	−3.7 (−13.5 to 6.2)	−0.191	0.467
	Post-intervention	87.3 ± 19.2	93.2 ± 20.4	−5.9 (−16.1 to 4.2)	−0.297	0.251
	MD (95% CI)	−15.6 (−18.6 to −12.7)	−17.9 (−22.3 to −13.6)			
	Effect size	−0.826	−0.890			
	*p*-value	0.001 *	0.001 *			
Quadriceps strength	Pre-intervention	146.6 ± 42.9	139.6 ± 41.2	7.0 (−4.4 to 28.6)	0.166	0.318
	Post-intervention	189.5 ± 48.9	174.7 ± 40.2	14.8 (−8.0 to 37.7)	0.330	0.200
	MD (95% CI)	−42.8 (−51.4 to −34.2)	−45.1 (−63.31 to −26.9)			
	Effect size	−0.932	−0.862			
	*p*-value	0.001 *	0.001 *			
Hip strength	Pre-intervention	0.5 ± 0.1	0.5 ± 0.1	0 (−0.1 to 0)	0	0.471
	Post-intervention	0.7 ± 0.1	0.7 ± 0.1	0 (−0.1 to 0)	0	0.700
	MD (95% CI)	−0.1 (−0.1 to 0)	−0.1 (−0.1 to 0)			
	Effect size	−1.999	−1.999			
	*p*-value	0.001 *	0.001 *			
Hamstring AT	Pre-intervention	93.3 ± 21.3	88.3 ± 18.2	5.0 (−5.2 to 15.1)	0.252	0.335
	Post-intervention	72.6 ± 20.1	66.4 ± 19.9	6.2 (−4.0 to 16.4)	0.309	0.231
	MD (95% CI)	20.6 (10.4 to 30.9)	21.9 (15.5 to 28.3)			
	Effect size	0.999	1.148			
	*p*-value	0.001 *	0.001 *			
Quadriceps AT	Pre-intervention	99.3 ± 25.7	92.9 ± 24.9	6.4 (−6.5 to 19.4)	0.252	0.325
	Post-intervention	64.0 ± 19.0	56.4 ± 18.5	7.6 (−2.0 to 17.1)	0.405	0.122
	MD (95% CI)	35.3 (26.4 to 44.1)	36.4 (28.0 to 44.8)			
	Effect size	1.561	1.664			
	*p*-value	0.001 *	0.001 *			

Supervised-rehab, supervised rehabilitation; Tele-rehab, telerehabilitation; AT, acceleration time; MD, mean difference; CI, confidence interval. Note: Values are expressed as mean ± standard deviation. The measurement unit of knee muscle strength was Nm kg^−1^ × 100. The measurement unit of hip muscle strength was kg-f. The measurement unit of acceleration time was milliseconds. All data were recorded and described by one physical therapist. * Statistically significant.

**Table 3 ijerph-20-02233-t003:** Comparison of patient-reported outcomes in the supervised-rehab and tele-rehab groups.

		Supervised-Rehab Group	Tele-Rehab Group	MD (95% CI)	Effect Size	*p*-Value
VAS score	Pre-intervention	4.7 ± 0.7	4.8 ± 0.7	−0.1 (−3.8 to 3.4)	−0.142	0.903
	Post-intervention	2.5 ± 0.6	2.4 ± 0.9	1.1 (−2.9 to 5.4)	0.130	0.547
	MD (95% CI)	21.5 (18.7 to 24.4)	23.0 (19.5 to 26.5)			
	Effect size	3.374	2.976			
	*p*-value	0.001 *	0.001 *			
AKPS score	Pre-intervention	65.7 ± 3.9	63.9 ± 7.9	1.8 (−1.4 to 5.0)	0.288	0.262
	Post-intervention	77.5 ± 4.3	78.9 ± 6.4	−1.4 (−4.3 to 1.3)	−0.256	0.304
	MD (95% CI)	−11.7 (−13.8 to −9.6)	−15.0 (−18.2 to −11.8)			
	Effect size	2.874	−2.086			
	*p*-value	0.001 *	0.001 *			
TSK-11 score	Pre-intervention	32.4 ± 7.2	33.3 ± 3.7	−0.9 (−3.9 to 1.9)	−0.157	0.503
	Post-intervention	21.1 ± 2.7	20.7 ± 3.6	0.4 (−1.2 to 2.0)	0.125	0.638
	MD (95% CI)	11.3 (8.7 to 13.8)	12.6 (11.0 to 14.2)			
	Effect size	2.078	3.451			
	*p*-value	0.001 *	0.001 *			

Supervised-rehab, supervised rehabilitation; Tele-rehab, telerehabilitation; VAS, visual analog scale; AKPS, anterior knee pain scale; TSK-11, tampa scale for kinesiophobia; MD, mean difference; CI, confidence interval. Note: Values are expressed as mean ± standard deviation. All data were recorded and described by one physical therapist. * Statistically significant.

**Table 4 ijerph-20-02233-t004:** Correlations between adherence rate and rate of change in muscle strength, muscle activation time, and patient-reported outcomes.

Parameters		Supervised-Rehab Group	Tele-Rehab Group
		Adherence Rate	Adherence Rate
Hamstring strength	PCC (r)	−0.111	0.174
	*p*-value	0.559	0.350
Quadriceps strength	PCC (r)	0.516	0.637
	*p*-value	0.002 *	0.001 *
Hip strength	PCC (r)	0.494	−0.086
	*p*-value	0.006 *	0.644
Hamstring AT	PCC (r)	−0.283	−0.266
	*p*-value	0.129	0.148
Quadriceps AT	PCC (r)	0.432	0.371
	*p*-value	0.021 *	0.040 *
VAS score	PCC (r)	0.091	0.368
	*p*-value	0.634	0.042 *
AKPS score	PCC (r)	0.650	0.456
	*p*-value	0.001 *	0.010 *
TSK-11 score	PCC (r)	0.444	0.509
	*p*-value	0.014 *	0.003 *

Supervised-rehab, supervised rehabilitation; Tele-rehab, telerehabilitation; AT, acceleration time; VAS, visual analog scale; AKPS, anterior knee pain scale; TSK-11, tampa scale for kinesiophobia; PCC, Pearson’s correlation coefficient. * Statistically significant.

## Data Availability

The data presented in this study are available on request from the corresponding author.

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
