# Peer review of "Comparison of Functional Outcomes between Supervised Rehabilitation and Telerehabilitation in Female Patients with Patellofemoral Pain Syndrome during the COVID-19 Pandemic"

_ijerph, 2023, doi:10.3390/ijerph20032233_

Round 1

Reviewer 1 Report

In this manuscript, a study is presented in which supervised rehabilitation and telerehabilitation were compared as treatments for female patients with patellofemoral pain syndrome. The topic falls squarely within the domain covered by this journal. The prospective longitudinal research design and inclusion of multiple types of outcome measures are desirable features of the manuscript. These positive impressions notwithstanding, I have several concerns about the manuscript in its current form:

1.         The null hypothesis is not a strong prediction. A more nuanced way of describing the expected results might be to state that effect sizes of similar magnitudes were anticipated for both treatments.

2.         Nonadherent patients should not be removed from the study because, as alluded to in the Discussion, poor adherence may be more likely to occur in telerehabilitation than in supervised rehabilitation. At a minimum, “intention to treat” analyses that include the deleted participants should be performed and reported alongside the analyses currently in the manuscript.

3.         The two groups should be compared on adherence levels.  Also, how adherence levels were measured should be mentioned and associations between adherence and study outcomes should be reported.

4.         Were participants randomly assigned to conditions? If not, that should be noted as a limitation of the study and the study should be described as “quasi-experimental” so as not to mislead the readership into thinking that the study is a randomized trial. 

Author Response

Detailed response to the reviewers

Dear editors and reviewers,

We deeply appreciate your thoughtful review and comments on our manuscript (Manuscript ID: IJERPH-2163021 entitled "Comparison of functional outcomes between supervised rehabilitation and telerehabilitation in female patients with patellofemoral pain syndrome during the COVID-19 pandemic").

We have reviewed the comments carefully and revised our manuscript accordingly.

The reviewer’s comments and questions have been addressed in a point-by-point manner.

Thank you again for your consideration. Please see attached file

Sincerely,

Ki-Mo Jang, MD, PhD.

Reviewer 2 Report

Thank you very much for inviting me to revier the paper entitled- Comparison of functional outcomes between supervised rehabilitation and telerehabilitation in female patients with patellofemoral pain synadrome during the COVID-19 pandemic

General comments: A well-written and scientifically interesting manuscript that addresses an important area of medical need. The authors was to give answers to important questions about the treatment with therapeutic exercises in women with anterior knee pain, and the outcomes between supervised rehabilitation and telerehabilitation, which makes it a meritorius manuscript.

Please see my specific comments below for more details,

This manuscript consists of a non-structured abstract with 6 keywords, 5 sections (introduction, materials & methods with 6 subsections, results with 3 subsections, discussion with limitations of the study, and conclusions) on 11 pages of single-spaced text with embedded figures (2) and tables (3). There are 51 references.

Specific comments:

  1. The keywords are absolutely fine.
  2. Introduction: It would be posible to add more information in this section, regarding the treatment in rehabilitation for patellofemoral pain syndrome (PFPS) ?, Why therapeutic exercise alone and not combined with another techniques…?

Line 57-59…could develop this information?, does the beginning of early rehabilitation increase the risk of complications? Is there evidence of how it is carried out in other countries, with good or bad results?

  1. Materials and methods: The methodology is rigorous. It is posible to add the brochure with pictures of exercises program in the manuscript?

4.      Results: Regarding the results, I sugest the authors, if the content in  Tables 1,2 and 3, is the same as in the lines (195-215) of the manuscript, perhaps  I would try to reduce numerical data in text and increase in tables, since the current format it makes difficult to read and understand.

I read “In this study… “ 7 times (lines: 72, 79, 94,97,102,225, 240…) please, is it posible to use some other synonym? Like “In the present study…, In our study”… or similar.

5.      The bibliography is current and encompasses the most recent scientific advances for research.

Thanks again for the invitation.

Author Response

(The authors gave the same response as above.)

Round 2

Reviewer 1 Report

I appreciate the efforts of the authors in addressing my concerns.  The quality of the manuscript has been greatly improved as a result of the changes that have been made.  The manuscript will make a valuable contribution to the literature.